# The Role of l-Carnitine in Mitochondria, Prevention of Metabolic Inflexibility and Disease Initiation

**DOI:** 10.3390/ijms23052717

**Published:** 2022-02-28

**Authors:** Mohamed Ashraf Virmani, Maria Cirulli

**Affiliations:** Alfasigma, 3528 BG Utrecht, The Netherlands; maria.cirulli@alfasigma.com

**Keywords:** mitochondrial function, l-carnitine, fatty acid oxidation, glycolysis, ketogenesis, beta oxidation, metabolic inflexibility, diabetes, neurodegeneration, liver disease

## Abstract

Mitochondria control cellular fate by various mechanisms and are key drivers of cellular metabolism. Although the main function of mitochondria is energy production, they are also involved in cellular detoxification, cellular stabilization, as well as control of ketogenesis and glucogenesis. Conditions like neurodegenerative disease, insulin resistance, endocrine imbalances, liver and kidney disease are intimately linked to metabolic disorders or inflexibility and to mitochondrial dysfunction. Mitochondrial dysfunction due to a relative lack of micronutrients and substrates is implicated in the development of many chronic diseases. l-carnitine is one of the key nutrients for proper mitochondrial function and is notable for its role in fatty acid oxidation. l-carnitine also plays a major part in protecting cellular membranes, preventing fatty acid accumulation, modulating ketogenesis and glucogenesis and in the elimination of toxic metabolites. l-carnitine deficiency has been observed in many diseases including organic acidurias, inborn errors of metabolism, endocrine imbalances, liver and kidney disease. The protective effects of micronutrients targeting mitochondria hold considerable promise for the management of age and metabolic related diseases. Preventing nutrient deficiencies like l-carnitine can be beneficial in maintaining metabolic flexibility via the optimization of mitochondrial function. This paper reviews the critical role of l-carnitine in mitochondrial function, metabolic flexibility and in other pathophysiological cellular mechanisms.

## 1. Introduction

l-carnitine is a vital molecule that is found in all living cells. It is a quaternary amine (3-hydroxy-4-*N*-trimethylaminobutyrate) whose main function in mammalian cells is the transfer of long chain fatty acids across the inner mitochondrial membrane for β- oxidation and generation of ATP energy. This process requires specific enzymes and transporters and any defects in these can cause disorders of the carnitine cycle. Once fatty acids are inside mitochondria, they undergo the process of β-oxidation through a series of acyl-CoA dehydrogenase enzymes that mediate the shortening of long-chain fats used to produce ATP. Deficiency of any of these enzymes can cause symptoms similar to those encountered in disorders of the carnitine cycle.

l-carnitine is also involved in the excretion of unwanted products of intermediary metabolism and in balancing the Coenzyme A (CoA) pool within mitochondria. Its unique properties enable other functions in cell metabolism such as buffering excess acyl residues and removal of xenobiotics from cells. These actions give l-carnitine a role in protecting cellular membranes, preventing fatty acid accumulation, modulating ketogenesis and glucogenesis and elimination of toxic metabolites.

## 2. l-Carnitine, Mitochondria and Cellular Metabolism

Metabolism is a coordinated series of chemical reactions that occur in cells of living organisms, needed to create and sustain life. These biochemical processes are important for the storage, freeing, and use of energy, and for the proliferation and repair of cells. The rate of recovery and the overall health of the body are therefore intrinsically linked to the capabilities of these metabolic processes.

Most of the cellular energy production comes from mitochondria whose main function is the production of adenosine triphosphate (ATP), which is needed to drive cellular processes (Figure 1).

Mitochondrial dysfunction leads to a loss of energy production, fatty acid accumulation, excess reactive oxygen species(ROS) generation, membrane instability, and a shift to glucose metabolism [1,2].

l-carnitine is central to mitochondrial processes, fatty acid metabolism, and the production of ATP. It also plays a major part in other important metabolic functions like cellular detoxification, control of ketogenesis and glucogenesis, and stabilization of cell membranes (Table 1). New research suggests that it also participates in blood glucose homeostasis [3]. l-carnitine continues to receive scientific attention as a therapy for kidney disease, cardiovascular disease, and diabetes, as well as symptoms associated with carnitine deficiency and mitochondrial disorders [4,5,6].

### 2.1. l-Carnitine and Fatty Acid Metabolism

l-carnitine plays a fundamental part in transporting long-chain fatty acids across the inner mitochondrial membrane since it is impermeable to fatty acids with carbon chain lengths greater than 12 [11]. l-carnitine is the only molecule capable of carrying fatty acids across the inner membrane and into the mitochondria, where they undergo β-oxidation. There is a complex system of enzymes and transporters that permit the l-carnitine molecule to enter the cell and to transfer fatty acid molecules into the mitochondria (Figure 1 and Figure 2).

Beta-oxidation produces acetyl fragments from the fatty acid molecule in the form of acetyl-Coenzyme A (acetyl CoA), which enters the Krebs (Citric acid) cycle and participates in the formation of ATP energy. The long chain fatty acids first bind to CoA to form the fatty acyl CoA that in turn binds with l-carnitine, and the resulting acylcarnitine is shuttled through the inner membrane and into the mitochondria (Figure 2). The acylcarnitine undergoes the reverse process in the matrix liberating the fatty acyl CoA for their subsequent β-oxidation. The breakdown of the fatty acid molecule by β-oxidation into smaller two carbon molecules permit their entry into the Krebs cycle and subsequent production of ATP [7,12].

### 2.2. Regulation of the Mitochondrial Acetyl-CoA/CoA Ratio and Acyl-CoA/CoA Ratio

l-carnitine has a key role in the regulation of acyl-CoA and acetyl-CoA, and in the preservation of free CoA levels within the mitochondria, which is crucial for maintaining metabolic flexibility.

l-carnitine plays a part in the carrier of activated acetyl and acyl groups. In partnership with the carnitine transferase enzyme (CPT) and carnitine translocase (CACT), l-carnitine forms an effective transport system for acetyl groups (Figure 3) or acyl groups (Figure 7) out of the mitochondria, thus preventing diminution of intramitochondrial free CoA. The buffering of the acetyl-CoA and acyl-CoA levels is important in allowing CoA-dependent reactions, especially glycolysis and the pyruvate oxidation to continue optimally [2,13].

The CACT translocase and CAT carnitine transferase, like all enzymes, can work in the reverse direction depending on the conditions. In this way, activated short, medium and long-chain acyl groups can be transported out of the mitochondrial matrix. In certain situations, cellular acyl-CoAs build-up as a result of an inborn error of fatty acid metabolism, and this also leads to an increase in acylcarnitines within the cell [12,14,15].

### 2.3. Detoxification of Toxic Metabolites

The build-up of potentially toxic metabolites in the cell due to mitochondrial dysfunction and other mechanisms can also alter metabolic flexibility. l-carnitine plays a part in removing potentially toxic metabolites associated with the β-oxidation of fatty acids (Figure 7). l-carnitine binds acyl residues arising from the intermediary metabolism of amino acids and helps in their elimination [16]. This mechanism is also essential in sequestering and subsequent removal of abnormal organic acids in several organic acidemia [8].

l-carnitine coupling decreases the number of long chain acyl residues attached to CoA and increases the free CoA to acyl-CoA’s ratio [9]. This process is essential in disorders of mitochondrial fatty acids oxidation where the fatty acyl-CoAs accrual in specific tissues such as in the heart and central nervous system can induce toxicity, trigger apoptosis and inflammation, and destabilize mitochondrial membrane [17,18]. For this reason, the conjugation of specific chain length acyls with l-carnitine creates very particular species of acylcarnitines that reflect any inborn errors of metabolism; thereby, they can also be utilized as markers for the screening of newborns [8,10].

### 2.4. Stabilization of Cell Membranes

l-carnitine is also essential in maintaining membrane stability and function of plasma, mitochondria, and other organelles possibly via effects on acetylation of membrane phospholipids. Its amphiphilic nature would also allow for interaction with the surface charges on the cell membrane and may play a role in membrane stabilization [19,20]. The charged tri-methylamino group and the carboxylic group on l-carnitine would permit interaction with corresponding poles on the membrane phospholipids, glycolipids, and proteins (Figure 4).

### 2.5. Control of Ketogenesis and Gluconeogenesis

Depending on the body state, e.g., fed or unfed, the liver regulates glucose and fatty acid synthesis, uptake, or release (Figure 5). In the postprandial (fed) state, the liver produces glucose, stops fatty acid synthesis, starts fatty acid oxidation, and produces ketone bodies [21].

During the catabolic state, the liver is a net producer of glucose by both glycogen breakdown and gluconeogenesis with more than half of the glucose produced being used by the brain to produce energy. Fatty acids from triglycerides are converted to ketones in the liver and are used as the second readily available fuel by the brain.

Ketogenesis occurs primarily in the mitochondria of liver cells, and CPT-1 is the rate limiting enzyme in this process. Fatty acids brought into the mitochondria via CPT-1 are broken down into acetyl CoA via β-oxidation. Two acetyl-CoA molecules are converted into acetoacetyl-CoA, which is then converted to HMG-CoA and subsequently to acetoacetate. Once acetoacetate reaches extrahepatic tissues, it is converted back to acetyl-CoA that can enter the citric acid cycle to produce ATP.

Ketogenesis is mainly modulated by insulin but also by other hormones like glucagon, cortisol, thyroid hormones, and catecholamines, which can increase free fatty acids availability for the ketogenic pathway [21,22].

Mitochondria are also involved in insulin signaling. A number of key enzymes in the ketogenic pathway are activated when insulin levels are low. This low insulin state produces an increase in free fatty acids that are taken up by the mitochondria for the consequent increase in ketone body synthesis.

l-carnitine affects ketogenesis via its part in the uptake of free fatty acids and their subsequent use in the production of ketones. Ketone body production is affected by l-carnitine status. In carnitine deficient states, long-chain fatty acyl-CoAs cannot be efficiently transported into mitochondria, thereby limiting the ketone body production [23].

## 3. Metabolic Inflexibility

Metabolic flexibility is the capacity of the body to use different energy sources depending on the circumstances and availability of oxygen, substrates, and nutrient cofactors. The inability to adapt to the different cellular and environmental needs/stressors results in metabolic inflexibility, which ultimately leads to disease states like insulin resistance, lipid accumulation, and an inability to maintain homeostatic balance [24,25,26] (Figure 6).

The onset of metabolic inflexibility has been linked to micronutrient deficiencies and nutrient overload, as well as the inevitable aging process. Recent research suggests that this process can be delayed or even reversed. To maintain metabolic flexibility, mitochondria need substrates, oxygen, and cofactors for energy production. Altered mitochondrial function leads to a reduction in energy production and an increase in reactive oxygen species, which negatively impacts all cellular functions such as calcium handling, anaplerosis, proteostasis, apoptosis, and autophagy [27,28,29].

An imbalanced diet with a high intake of processed foods and fat leads to the phenomenon known as “empty calories”. Although foods with empty calories provide immediate energy, they cannot be used efficiently for maintaining body homeostasis and are ultimately stored as fat. Chronic overnutrition triggers mitochondrial metabolic confusion and reduced metabolic efficiency. This results in the accumulation of electrons which can trigger ROS generation, as well as an accumulation of acyl-CoA which further disrupts mitochondrial function and signaling. An inefficient mitochondrial buffering and repair system can ultimately lead to cellular dysfunction [24,27].

Overeating and lack of exercise can lead to an excess of energy intake relative to body requirements. This can cause a malfunctioning of body control systems altering the delicate balance between requirements and expenditure. The systems controlling the body’s energy requirements are complex and they involve integration of metabolic and neuroendocrine signals. This occurs mainly in the brain and in particular in the hypothalamus; however, it also involves sensors at the cellular and organelle level in the liver, pancreas, muscles, adipose tissues, and gut [30,31]. Mitochondrial dynamics also play an important role in energy homeostasis due to their ATP producing capacity [32].

Mitochondrial dysfunction may underly cellular senescence when cells accumulate damage and lose their metabolic flexibility [33,34]. When cellular metabolism is efficient, senescent cells are removed from the body by apoptosis. If these cells are not eliminated, it can result in disease states like cancer, kidney, liver, and heart disease.

Together with body control systems, mitochondria are key players in maintaining energy requirements and cell function. This will ensure a healthy body homeostasis and prevent the initiation of disease processes. To maintain metabolic energy equilibrium, it is necessary to eat the required type and amount of food, avoid eating empty calorie foods, and ensure correct micronutrient intake. Improving mitochondria function with a balanced diet and lifestyle but also correcting nutrient deficiencies can play a crucial role in maintaining metabolic flexibility and health [35,36].

## 4. The Role of l-Carnitine in Improving Metabolic Flexibility

### 4.1. Metabolic Flexibility and Acetyl-CoA/CoA Ratio

The presence of free carnitine within the mitochondria allows for the generation of acetyl-l-carnitine (ALC) necessary for the reduction of acetyl-CoA and the generation of free CoA which is needed for glycolysis to proceed. (Figure 3). The consumption of free l-carnitine and free CoA due to the accumulation of long chain acyl groups can limit acetyl-l-carnitine synthesis, since free l-carnitine levels are critical for maintaining the buffering capacity of CPT’s. Therefore, low CPT activity and diminished l-carnitine levels may contribute to the development of mitochondrial metabolic inflexibility [37,38,39,40].

Buffering of the long chain acyl groups and maintaining optimal levels of acetyl and free CoA pools is crucial in conditions that can affect the mitochondrial substrate availability, for instance in overfeeding and exercise. High levels of acetyl-CoA and nutrient overabundance with a lack of specific nutrient cofactors can result in contradictory signals managing mitochondrial substrate flow and usage, and therefore metabolic inflexibility.

### 4.2. Metabolic Flexibility and Acyl-CoA/CoA Ratio

Disease states like kidney disease, fatty liver disease, insulin resistance and neurodegenerative diseases, myalgic encephalomyelitis/chronic fatigue syndrome, and fertility disorders may be linked to metabolic inflexibility caused by alterations in the carnitine cycle. In fact, l-carnitine deficiencies and elevated long-chain acylcarnitines are seen in many of these disease states [1,41,42,43].

A number of enzymes are inhibited by high levels of acyl-CoA compounds, including acetyl-CoA carboxylase, adenine nucleotide translocase, and pyruvate dehydrogenase and can contribute to metabolic inflexibility. l-carnitine supports mitochondrial function by preventing the accumulation of long chain fatty acyl groups, removing potentially toxic acyl compounds, and maintaining the acyl-CoA/CoA ratio (Figure 7) [10,20]. This l-carnitine buffering effect also extends to the potentially toxic acyl-groups, resulting either from poorly metabolized fatty acyls and from xenobiotics such as antibiotics conjugated with pivalic acid and valproate [44], or from inhibition of certain metabolic pathways for example propionic acid in propionic acidemia [45]. These potentially toxic acyl-groups are converted to CoA-derivatives, thereby exhausting free CoA pools. The consequent binding of these acyl-CoAs to l-carnitine forming the acylcarnitines allows their exit from the mitochondria, the cell, and for eventual excretion in the urine. This process also increases the free CoA pool, but since it uses l-carnitine, it can cause a deficiency that can be countered by the supplementation of l-carnitine [46,47,48].

## 5. Role of Carnitine in Disease

### 5.1. l-Carnitine and Insulin Resistance

Insulin resistance is in part related to reduced metabolic flexibility. Clinical and preclinical studies have shown that “lipid over supply” causes or worsens insulin resistance via multiple mechanisms including the accumulation of intracellular lipids in various tissues. The build-up of fatty acyl CoA derivatives is known to inhibit insulin signaling and glucose oxidation in muscle [1]. As mentioned previously, l-carnitine buffers the long chain acyl groups which are bound to CoA, as well as the high acetyl CoA within the mitochondria, and frees up CoA, which is crucial for metabolism to proceed (Figure 3 and Figure 7).

Lipid oversupply to the mitochondria exceeds the capacity of the enzymatic pathways, leading to an imbalance between β-oxidation and the Krebs cycle. This results in an increase of long chain acyl-CoAs, as well as the long chain acylcarnitines. In fact, high levels of long chain acylcarnitines are seen in insulin resistance and reduced metabolic flexibility [37,42].

l-carnitine has been shown to improve glucose tolerance and insulin sensitivity. A number of mechanisms have been proposed that may underlie this favorable effect on glucose metabolism by carnitine [49,50,51,52]. These include:Improved mitochondrial oxidation of the long chain acyl CoA since their accretion is linked to insulin resistance;Increasing the intramitochondrial acetyl-CoA/CoA which is positive for pyruvate dehydrogenase complex (PDHC) activity;Improving expression of enzymes in the glycolytic and gluconeogenic;Improved expression of genes in the insulin signaling cascade;Improved signaling cascade for insulin-like growth factor-1 (IGF-1) axis and IGF-1.

These metabolic actions suggest that carnitine supplementation may improve metabolic flexibility and reduce insulin resistance.

### 5.2. l-Carnitine and Endocrine Imbalances

Mitochondria are increasingly being identified for their critical role in fertility. Sperm cells and oocytes are very rich in mitochondria and have high energy needs [53,54,55,56,57]. High energy is needed for the different reproductive stages like sperm and oocyte maturation, ovulation, implantation, and pregnancy. High levels of carnitine are found in fertility related tissues like the testis, ovary, and epididymis, underlying the importance of l-carnitine in the reproductive system.

Mitochondrial function plays a crucial part in sperm and oocyte quality and is important for successful reproduction. The reproductive organs in common with most other tissues of the body have metabolic and energy requirements that change during development and through adult function. The largest cell in the body is the mature oocyte, and this cell has a high mitochondrial content. As the oocyte develops from the primordial germ cells the mitochondrial numbers and their functions change. Since these organelles are not capable of de novo biogenesis new ones are generated by expansion and fission of the existing mitochondria with the cell. In the fertilized mature oocyte, only the maternal mitochondrial DNA is kept with the mitochondria from the sperm being actively excluded by the zygote [58].

Low levels of l-carnitine are seen in men with oligoasthenospermia, in women with polycystic ovary syndrome (PCOS), and in ovarian hyperstimulation [59]. l-carnitine treatment has been shown to be effective in women undergoing superovulation procedures for IVF, in men preparing for ICSI, and in women with PCOS [43,60]. l-carnitine has also been shown to improve the lipid profile and insulin sensitivity in women with PCOS [61,62,63,64,65].

The neuroendocrine system is closely linked to l-carnitine synthesis and utilization, and the release of GnRH by the hypothalamus is improved by supplementation with l-carnitine [66]. Preclinical studies demonstrated that l-carnitine improves β-oxidation in mitochondria resulting in improved fertilization rate and blastocyte development [67,68,69].

The beneficial role of l-carnitine in fertility related conditions could be linked to its role in regulating mitochondrial function and metabolic flexibility resulting in an improved energy production, as well as neuroendocrine signaling.

### 5.3. l-Carnitine and Fatty Liver Disease

Fatty liver disease, also known as steatosis, develops when there is a disparity between fatty acid uptake, synthesis, and disposal from the liver of very low-density lipoprotein (VLDL) and fatty acid oxidation. Numerous studies have shown that l-carnitine administration can improve or prevent liver damage due to various insults by ameliorating hepatic mitochondria β-oxidation and reducing oxidative stress. l-carnitine was shown to increase gene expression in the liver related to fatty acid transport, β-oxidation, and antioxidant enzymes [70].

Carnitine deficiency seen in liver disease may cause metabolic dysfunctions in gluconeogenesis, fatty acid metabolism, albumin biosynthesis, and ammonia detoxification by the urea cycle [71].

l-carnitine administration has also been shown to improve markers of glycemic control in patients with non-alcoholic fatty liver disease (NAFLD) and diabetes, most likely by regulating the ratio of acetyl-CoA/CoA in the mitochondria and thereby the PDH flux [72]. l-carnitine supplementation can modulate insulin sensitivity and glucose uptake and have an antioxidant effect on hepatocytes. l-carnitine also ameliorates systemic inflammation, mitigates ROS production, and prevents fibrosis progression by acting on intra- and extracellular key signaling molecules.

The liver is the main site for l-carnitine synthesis [73]. Reduced levels of l-carnitine often seen in patients with liver disease may negatively impact fatty acid oxidation, increase ROS production, and contribute to mitochondrial impairment [70,71]. The link between carnitine and liver disease is evidenced by the fact that patients with liver disease share similarities with primary carnitine deficiency (PCD) patients. PCD is an autosomal recessive disorder involving compromised fatty acid oxidation caused by l-carnitine deficiency due to a lack of its OCTN transporter [8,15,16].

In patients with PCD, the low serum and intracellular levels of l-carnitine lead to impairment in fatty acids utilization and their accumulation in the body. Consequently, metabolic inflexibility occurs with the body relying exclusively on glucose as the energy source and fat accumulates in the liver. Patients with PCD are at risk of many complications, especially encephalopathy that is a major problem in advanced liver disease. Furthermore, elevated levels of liver enzymes like alanine transaminase (ALT) and aspartate transaminase (AST) are observed in both PCD and fatty liver disease, reflecting the underlying liver damage. Inhibiting CPT1 with the drug etomoxir causes the switch of energy metabolism from fatty acid to glucose oxidation and is linked with severe hepatotoxicity [74,75].

A relative lack of l-carnitine leads to reduced fatty acid oxidation and triglyceride accumulation in the liver and supplementation with l-carnitine can improve mitochondrial function and metabolic flexibility.

### 5.4. l-Carnitine in Neurodegenerative Diseases

Mitochondrial dysfunction appears to be an important early underlying factor in the development of neurodegenerative diseases such as Alzheimer’s, Parkinson’s, Huntington’s, and dementia. Various preclinical models have been created that are showing the impact of metabolism on brain function and in particular the role that impaired metabolism plays in the neurodegeneration processes. The major pathways for the central nervous system (CNS) toxicity could be related to increased neuronal and glial injury due to metabolic impairment linked to mitochondrial dysfunction. This would result in:Augmented excitotoxicity;Diminished energy production;Reduced antioxidant potential.

Mitochondrial dysfunction can also affect neuronal membrane composition and influence all cellular processes like gene response, epigenetic programming, and inflammatory pathways [76,77].

Cell death is probably attributed to lack of energy, increased oxidative stress, and calcium overload in all cells, but it is especially critical to highly differentiated neurons. Inhibiting various mitochondrial enzymes, has shown that it can selectively kill neurons [78]. The resulting mitochondrial dysfunction can lead to neurotoxicity and development of neurodegenerative diseases.

Although the mechanism is not clear, it is believed that the build-up of amyloid beta protein plaques seen in the brain plays an important role in Alzheimer’s disease. Numerous studies have shown that the amyloid beta peptide exerts neurotoxic effects possibly via mitochondrial dysfunction [20,79,80,81].

l-carnitine and ALC have been linked to the prevention of toxic effects causes by beta amyloid (Aβ) and improve symptoms in Alzheimer’s disease. The neuroprotective effects of carnitines could be related to the reduction of amyloid related mitochondrial dysfunction and reduction in ROS levels [80,82]. Indeed, ALC was shown to reduce Aβ1-42-induced protein and lipid oxidation, as well as improve antioxidant potential by increasing glutathione and heat shock proteins [83] and by preventing ATP depletion induced by Aβ [84]. Experimental studies have shown that treating cortical neurons with ALC can counter the neurotoxic effects of amyloid Aβ25-35 fragment [80]. ALC also attenuates Aβ1-42 provoked toxicity and apoptosis which could be related to a decrease in protein and lipid oxidation [83].

A recent study in mature rat hippocampal neurons showed that l-carnitine and ALC protected the mitochondrial membrane potential and countered the decline in oxygen consumption rates and the rise in mitochondrial fragmentation provoked by Aβ1-42 [81]. ALC also ameliorated the Aβ-induced changes in mitochondrial movement, which may also be linked to decreased neuronal death by apoptosis.

In addition to carnitines, a number of metabolically active compounds like nicotinamide, thiamine, riboflavin, creatine, and coenzyme Q10 acting at the mitochondrial level have been shown to counteract neuronal damage in conditions of ischemia, hypoxia, and metabolic compromise [85,86].

### 5.5. Potential Use of l-Carnitine as a Therapeutic Agent

l-carnitine has been used as a therapeutic agent in many conditions associated with metabolic disorders and carnitine deficiency [87,88]. A deficiency of l-carnitine can lead to impaired mitochondrial function and cellular metabolic alterations that may underlie many disease states. Secondary l-carnitine deficiency in the body generally occurs due to malabsorption, defective biosynthesis, increased use or degradation, and defective kidney re-uptake. Inadequate intake may contribute to secondary carnitine deficiency, as may be seen in those receiving long-term parenteral nutrition without supplemental l-carnitine [89]. l-carnitine supplementation can help restore carnitine homeostasis and counteract altered metabolic alterations present in pre-disease and disease states.

Extensive preclinical and clinical research studies have confirmed the positive role that l-carnitine treatment has in male and female fertility [90,91,92,93], pregnancy, premature neonates [94,95,96]. liver disease [75,97], kidney disease [98,99], and in valproate-evoked toxicity [100,101]. l-carnitine treatment has also shown potential in the treatment of other conditions like myalgic encephalomyelitis (ME)/chronic fatigue syndrome (CFS) [102], neurodegenerative diseases [103], and diabetes [49,104,105]. New recent research is beginning to show that l-carnitine can also modulate gene expression and other critical biological processes, in addition to its key role in mitochondrial energy metabolism [5,106,107].

## 6. Summary—Disease Initiation and the Metabolic Approach

It is becoming apparent that the pathogenesis of conditions like neurodegeneration, insulin resistance, endocrine imbalances, liver and kidney disease are linked to metabolic disorders and in particular to mitochondrial dysfunction. Cellular functions and structures are an expression of metabolism and change according to it. The rate of recovery and the overall health of the body are intrinsically related to the proper functioning of metabolic processes. The inability to adapt to the different cellular and environmental needs or stressors results in metabolic inflexibility. The onset of metabolic inflexibility has been associated with micronutrient deficiencies and nutrient overload as well as the inevitable aging process. Proper mitochondrial function is central to the maintenance of metabolic flexibility.

Diet is known to have an important impact on health and disease. Mitochondrial dysfunction due to a relative lack of micronutrients and substrates like carnitines is implicated in the development of many chronic diseases and neurodegenerative diseases. To maintain metabolic flexibility, mitochondria need substrates, oxygen, and cofactors for energy production. Metabolic changes at the “micro” level (cellular functions, membranes, mitochondria) lead to “macro” alterations that impact the body as a whole and eventually manifest as age-related chronic and degenerative diseases. Disturbance in mitochondrial function may also contribute to or cause the fatigue seen in ME/CFS [102] and long-COVID [108].

Initially, the body tries to compensate for micronutrient and substrate deficiencies by taking the needed compounds from other areas of the body; for example, l-carnitine from the muscle and calcium from the bones. However, persistent nutrient deficiency can lead to a loss of metabolic function, and cellular damage. Research has shown that supplementing with endogenous and metabolic compounds can provide metabolic support that removes or minimizes the cause of cell damage.

l-carnitine is central to mitochondrial processes, fatty acid metabolism, and the production of ATP. l-carnitine also plays a critical role in other important metabolic functions like cellular detoxification, control of ketogenesis and glucogenesis, and stabilization of cell membranes. The protective effects of micronutrients like l-carnitine that target mitochondria are becoming better understood and hold considerable promise for the management of age and metabolic related diseases. Strategies to optimize mitochondrial function could play an important role in limiting disease initiation and progress. These could include diet, exercise, and the prevention of nutrient deficiencies which can all contribute to maintaining metabolic flexibility, energy equilibrium, and healthy body homeostasis.

## Figures and Tables

**Figure 1 ijms-23-02717-f001:**
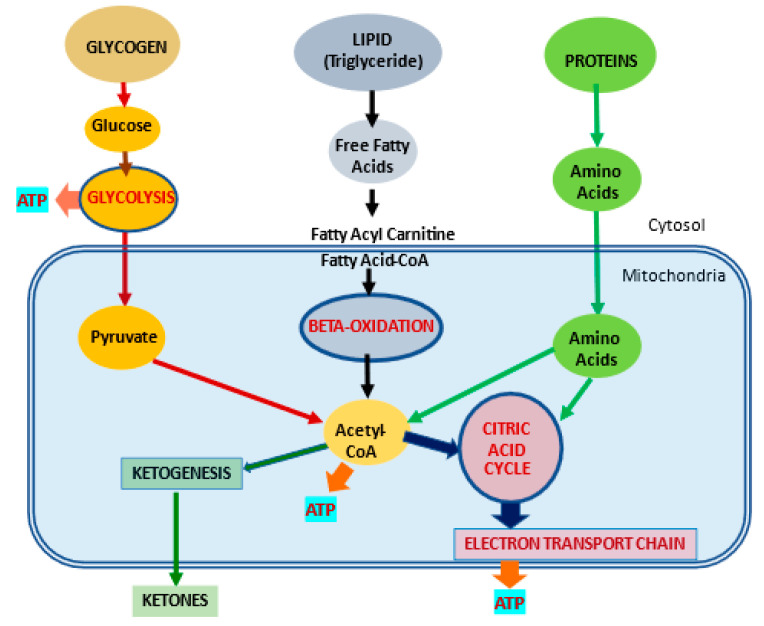
Metabolic pathways for major substrates in the body. Adapted from Virmani et al., 2015 [2]. The body utilizes carbohydrates as glucose, lipids as free fatty acids and proteins as amino acids as the major substrates for the production of ATP energy. Much of the energy from these substrates is produced in the mitochondria.

**Figure 2 ijms-23-02717-f002:**
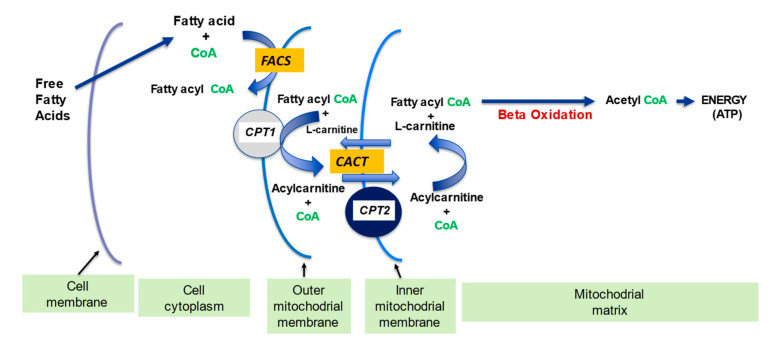
The Role of l-carnitine in fatty acid transport and β -oxidation in mitochondria. FACS: Acyl-CoA synthetase; CPT 1: carnitine palmitoyl transferase 1; CACT: carnitine acylcarnitine translocase; CPT 2: carnitine palmitoyl transferase 2. Adapted from Kerner and Hoppel 2000 [12]. The free fatty acids are first converted to fatty acyl CoA by the enzyme FACS, which is than attached to the l-carnitine by CPT1 to form the acylcarnitine, which is carried into the mitochondria by carrier CACT where it is converted back to fatty acyl CoA by CPT2.

**Figure 3 ijms-23-02717-f003:**
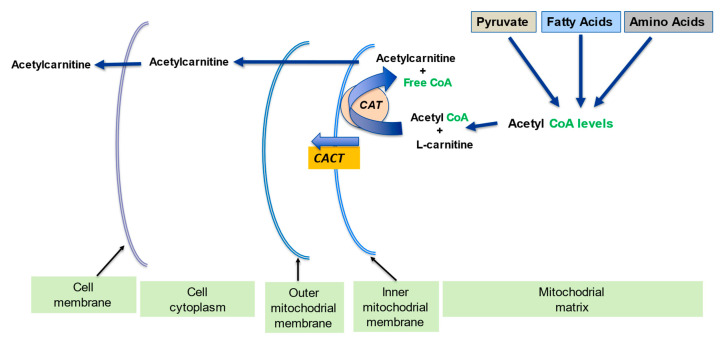
l-carnitine regulates the intramitochondrial acetyl-CoA/CoA ratio. Adapted from Virmani et al., 2015 [2]. The acetyl-CoA build up from pyruvate, fatty acids amino acids metabolism becomes a limiting step since free mitochondrial CoA levels decline. l-carnitine converts the acetyl-CoA to acetyl-l-carnitine, which can be transported out of the mitochondrial matrix, and in this way frees up the CoA.

**Figure 4 ijms-23-02717-f004:**
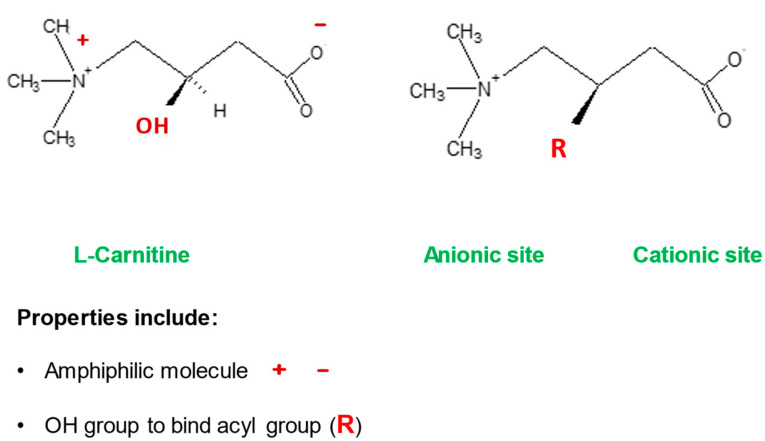
The amphiphilic nature of the l-carnitine molecule. Adapted from Virmani and Binienda, 2004 [20]. The unique amphiphilic nature of l-carnitine is attributed to the charges on its carboxyl and amine groups. The hydroxyl group gives it the ability to bind to fatty acids.

**Figure 5 ijms-23-02717-f005:**
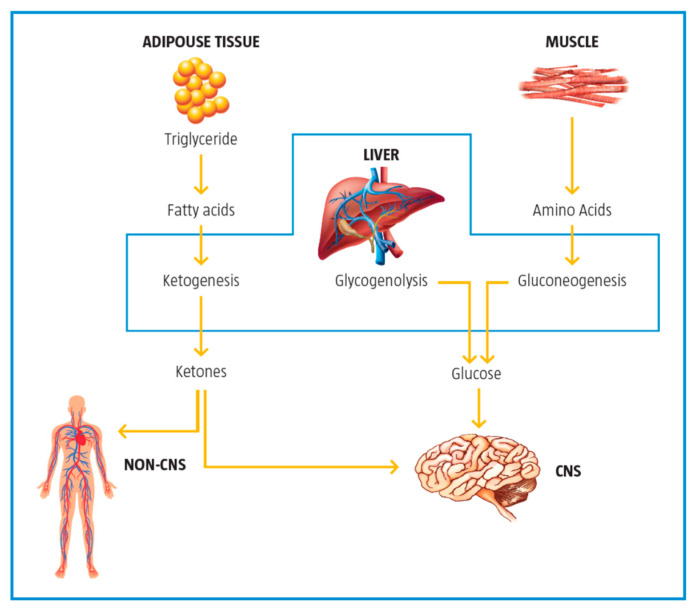
The role of the liver in glucose and fat metabolism. Adapted from Foster, 2004 [21].

**Figure 6 ijms-23-02717-f006:**
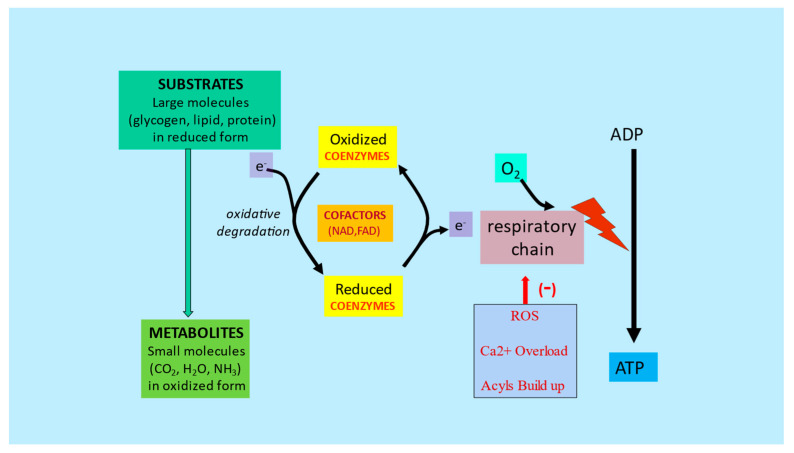
Substrates and nutrient cofactors in mitochondrial function and metabolic flexibility. Adapted from Virmani et al., 2013 [27]. Energy from starting substrates such as glycogen, lipids, and proteins are gradually extracted by the complex series of enzymes in the cellular cytoplasm and mitochondria, resulting in metabolites, mainly CO_2_ and H_2_O. Any dysfunction in this chain of events caused by lack of cofactors or mitochondrial dysfunction would lead to reduced ATP formation and increased ROS, Ca^2+^ and acyls buildup.

**Figure 7 ijms-23-02717-f007:**
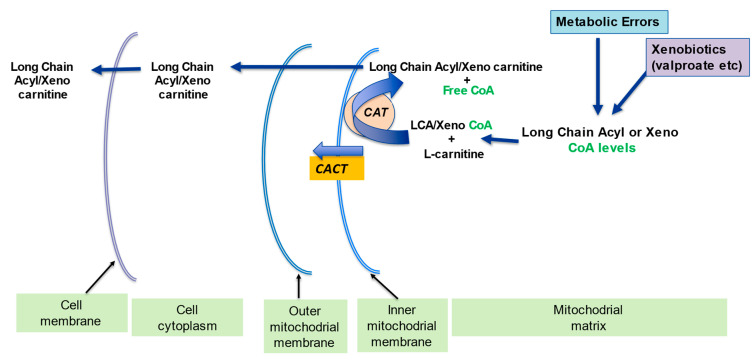
l-Carnitine needed for removal of long chain acyls and xenobiotics from mitochondria. Adapted from Virmani et al., 2015 [2]. The acetyl-CoA build-up of long chain acyl CoA and Xeno-CoA from metabolic errors and xenobiotics would reduce free mitochondrial CoA levels, as well as be detrimental by themselves. The l-carnitine coverts these long chain acyls-carnitine or Xeno-carnitine, which can be transported out of the mitochondrial matrix and the cell.

**Table 1 ijms-23-02717-t001:** Major l-carnitine functions in cellular metabolism.

l-Carnitine (LC) Function	Description	References
Fatty acid metabolism	LC transports LCFA across inner mitochondrial membrane for subsequent beta oxidation and ATP production	[4,5]
Regulation of mitochondrial acetyl-CoA/CoA ratio and acyl-CoA/CoA ratio	LC forms an effective transport system for acetyl or acyl groups out of the mitochondria to maintain free CoA levels important for glycolysis and other processes	[2,5]
Detoxification of potentially toxic metabolite	LC binds acyl residues and helps in their elimination thereby maintaining metabolic flexibility	[7]
Stabilization of cell membranes	LC helps stabilize cell membranes via its effects on acetylation of membrane phospholipids and surface membrane effects	[8,9]
Control of ketogenesis and gluconeogenesis	LC affects ketogenesis by transporting free fatty acids into mitochondria for subsequent use in the production of ketones in the mitochondria.	[10]

## Data Availability

Not applicable.

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
