# Peer review of "The Role of l-Carnitine in Mitochondria, Prevention of Metabolic Inflexibility and Disease Initiation"

_ijms, 2022, doi:10.3390/ijms23052717_

Round 1

Reviewer 1 Report

Title: The role of L-carnitine in mitochondria, prevention of metabolic inflexibility and disease initiation

Journal: International Journal of Molecular Sciences

In this article, Virmani et. al. have compiled a review article on the role of carnitine in mitochondrial metabolism. This is an interesting topic and the authors have done a tremendous job. However, revisions are required to tailor this article for this journal, and to improve the structure of the manuscript. Further, the manuscript should be thoroughly checked for English.

Please find my comments below:

  1. The manuscript should be thoroughly re-read again. Please check for sentences without appropriate citations. There are several sentences in the manuscript that are not supported by references. For example:
    1.  'New research suggests that it also plays a role in blood glucose homeostasis. L-carnitine continues to receive scientific attention as a therapy for kidney disease, cardiovascular disease, and diabetes as well as symptoms associated with carnitine deficiency, and mitochondrial disorders'
    2. 'L-carnitine plays a fundamental role in the transport of long-chain fatty acids across the inner mitochondrial membrane. This membrane is impermeable to long-chain fatty acids (≥C12) and L-carnitine is the only molecule capable of carrying fatty acids across the inner membrane and into the mitochondria, to then undergo beta-oxidation.'

  1. The authors can add an introduction section before a sub-section '1. L-carnitine, mitochondria and cellular metabolism'.  2-3 paragraphs or a single page on the overall biology of mitochondria, its role in metabolism the role of L-carnatine etc. can be added. It would gear the readers of subsequent expansion of these sections.
  2. The following sentences should be improved, restructured, and should be added with references. 'The body control systems for energy requirements are complex with sensors in the brain as well as hormonal and other sensory organs like the liver, pancreas, muscles etc. The energy sensory system also exists at the cellular level and organelle level especially the mitochondria.'

  1. To emphasize the function, and to improve the readability of the manuscript, authors can add a table with 4 columns. The headers can be, 1st column (name of the function:L-carnitine and fatty acid metabolism, Regulation of the mitochondrial acetyl-CoA/CoA ratio and acyl-CoA /CoA ratio, Detoxification of potentially toxic metabolite, Stabilization of cell membranes, Control of ketogenesis and gluconeogenesis ), 2nd column (one/two sentences on function), 3rd column with its role in disease initiation and 4th column (references).
  2. In one of instance authors have mentioned 'Numerous studies have shown that L-carnitine administration can ameliorate or prevent liver damage of various causes by ameliorating hepatic mitochondria β-oxidation and reducing oxidative stress [62,63,64].' The authors need to expand these findings in a couple of more sentences.
  3. The following sentences need appropriate references. 'The liver is the primary site for L-carnitine synthesis. Reduced levels of L-carnitine often seen in patients with liver disease may contribute to reduced fatty acid oxidation, increased ROS production and subsequent mitochondrial impairment. The link between carnitine and liver disease is evidenced by the fact that patients with liver disease share similarities with those with primary carnitine deficiency (PCD), an autosomal recessive disorder of fatty acid oxidation due to the lack of OCTN transporter.'
  4. The following sentence should be appropriately constructed and linked. 'Studies have shown that treatment of cortical neurons with ALC countered the toxicity of amyloid Aβ25-35 fragment [71]. and also attenuated Aβ1-42-induced cytotoxicity and apoptosis [74].'
  5. There is a lot of scope for improvement in the summary/concluding paragraphs. Authors should reconstruct key take-home points of the manuscript. The conclusion should crystallize the article and expand into future directions. The authors have compiled an interesting manuscript and if the authors read the manuscript again, the fine prints can be crystallized into a very good conclusion section.
  6. The authors can improve this sentence 'Cellular functions and structures are an expression of metabolism and change according to it. '

Author Response

Response in file attached

Reviewer 2 Report

Please find the review report attached.

Author Response

Response in file attached 

Round 2

Reviewer 1 Report

The authors have improved the manuscript, therefore, should be considered for publication.